# Multi-Defendant Legal Judgment Prediction via Hierarchical Reasoning

**Yougang Lyu**[1*]    **Jitai Hao**[1*]    **Zihan Wang**[1]    **Kai Zhao**[2]    **Shen Gao**[1]

**Pengjie Ren**[1]    **Zhumin Chen**[1]    **Fang Wang**[1]    **Zhaochun Ren**[3†]

[1]Shandong University, Qingdao, China
[2]Georgia State University, Atlanta, USA
[3]Leiden University, Leiden, The Netherlands
{youganglyu, 202215112, 202020630}@mail.sdu.edu.cn
kzhao4@gsu.edu, shengao@sdu.edu.cn, jay.ren@outlook.com
{chenzhumin, wangfang226}@sdu.edu.cn, z.ren@liacs.leidenuniv.nl

## Abstract

Multiple defendants in a criminal fact description generally exhibit complex interactions, and cannot be well handled by existing Legal Judgment Prediction (LJP) methods which focus on predicting judgment results (e.g., law articles, charges, and terms of penalty) for single-defendant cases. To address this problem, we propose the task of multi-defendant LJP, which aims to automatically predict the judgment results for each defendant of multi-defendant cases. Two challenges arise with the task of multi-defendant LJP: (1) indistinguishable judgment results among various defendants; and (2) the lack of a real-world dataset for training and evaluation. To tackle the first challenge, we formalize the multi-defendant judgment process as hierarchical reasoning chains and introduce a multi-defendant LJP method, named Hierarchical Reasoning Network (HRN), which follows the hierarchical reasoning chains to determine criminal relationships, sentencing circumstances, law articles, charges, and terms of penalty for each defendant. To tackle the second challenge, we collect a real-world multi-defendant LJP dataset, namely MultiLJP, to accelerate the relevant research in the future. Extensive experiments on MultiLJP verify the effectiveness of our proposed HRN.

## 1 Introduction

Legal Judgment Prediction (LJP) aims at predicting judgment results (e.g., law articles, applicable charges, and terms of penalty) based on the fact description of a given case. Existing LJP studies primarily focus on single-defendant cases, where only one defendant is involved (Luo et al., 2017; Zhong et al., 2018; Yang et al., 2019; Xu et al., 2020; Dong and Niu, 2021; Yue et al., 2021; Lyu et al., 2022; Feng et al., 2022b; Zhang et al., 2023a).

---

* Equal contribution.
† Corresponding author.

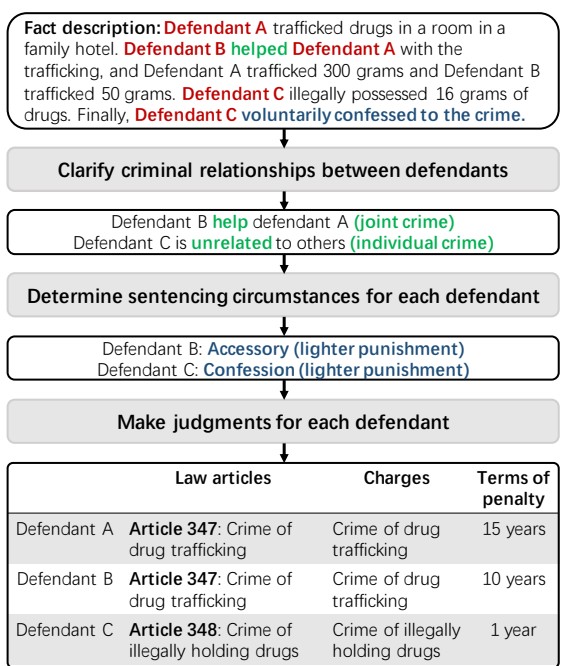

Figure 1: An illustration of multi-defendant LJP. Generally, a judge needs to reason on the fact description to clarify the complex interactions among different defendants and make accurate judgments for each defendant.

Despite these successful efforts, single-defendant LJP suffers from an inevitable restriction in practice: a large number of fact descriptions with multiple defendants. According to statistics derived from published legal documents sampled from legal case information disclosure, multi-defendant cases constitute a minimum of 30% of all cases (Pan et al., 2019). As shown in Figure 1, the multi-defendant LJP task aims at predicting law articles, charges, and terms of penalty for each defendant in multi-defendant cases. Since multiple defendants are mentioned in the fact description and exhibit complex interactions, the multi-defendant LJP task requires clarifying these interactions and making accurate judgments for each defendant, which is intuitively beyond the reach of single-defendant LJP methods. Hence, there is a pressing need to extend LJP from

single-defendant to multi-defendant scenarios.

However, two main challenges arise with the task of multi-defendant LJP:

- **Indistinguishable judgment results among various defendants**. As complicated interactions exist among multiple defendants, fact descriptions of various defendants are usually mixed together. Thus it is difficult to distinguish different judgment results among various defendants so as to make accurate judgments for each defendant. As shown in Figure 1, in order to distinguish different judgment results among various defendants, a judge has to clarify criminal relationships among defendants to determine whether defendants share same law articles and charges, and sentencing circumstances affecting terms of penalty for each defendant. Based on these intermediate reasoning results, the judge determines and verifies the judgment results (law articles, charges, and terms of penalty) for each defendant, following a forward and backward order. The motivation behind forward prediction and backward verification is rooted in the complex nature of legal reasoning, where evidence and conclusions can be interdependent (Zhong et al., 2018; Yang et al., 2019). Overall, the multi-defendant judgment process requires simulating the judicial logic of human judges and modeling complex reasoning chains.

- **Lack of real-world multi-defendant LJP datasets**. Existing datasets for LJP either only have single-defendant cases annotated for multiple LJP subtasks or multi-defendant cases for a single LJP subtask. Xiao et al. (2018) collect a real-world LJP dataset CAIL, which only retains single-defendant cases. Pan et al. (2019) only annotate multi-defendant cases with the charge prediction subtask and ignore information about criminal relationships and sentencing circumstances that can distinguish judgment results of multiple defendants in real scenarios. In order to accelerate the research on multi-defendant LJP, we urgently need a real-world multi-defendant LJP dataset.

To tackle the first challenge, we formalize the multi-defendant judgment process as hierarchical reasoning chains and propose a method for multi-defendant LJP, named Hierarchical Reasoning Network (HRN), which follows the hierarchical reasoning chains to distinguish different judgment results of various defendants. Specifically, the hi-

erarchical reasoning chains are divided into two levels. The first-level reasoning chain identifies the relationships between defendants and determines the sentencing circumstances for each defendant. The second-level reasoning chain predicts and verifies the law articles, charges, and terms of penalty for each defendant, using a forward prediction process and a backward verification process, respectively. Since generative language models have shown great ability to reason (Talmor et al., 2020; Yao et al., 2021; Hase and Bansal, 2021), we convert these reasoning chains into Sequence-to-Sequence (Seq2Seq) generation tasks and apply the mT5 (Xue et al., 2021) to model them. Furthermore, we adopt Fusion-in-Decoder (FID) (Izacard and Grave, 2021) to process multi-defendant fact descriptions with thousands of tokens efficiently.

To tackle the second challenge, we collect a real-world dataset, namely MultiLJP, with 23,717 real-world multi-defendant LJP cases. Eight professional annotators are involved in manually editing law articles, charges, terms of penalty, criminal relationships, and sentencing circumstances for each defendant. In 89.58 percent of these cases, the defendants have different judgment results for at least one of the subtasks of the multi-defendant LJP task. MultiLJP requires accurate distinction of the judgment results for each defendant. This makes MultiLJP different from the existing single-defendant LJP datasets. Our work provides the first benchmark for the multi-defendant LJP task.

Using MultiLJP, we evaluate the effectiveness of HRN for multi-defendant LJP on various subtasks. The results show that HRN can significantly outperform all the baselines. In summary, our main contributions are:

- We focus on the multi-defendant LJP task and formalize the multi-defendant judgment process as hierarchical reasoning chains for the multi-defendant LJP task.
- We introduce HRN, a novel method that follows the hierarchical reasoning chains to distinguish the judgment results for each defendant in multi-defendant LJP.
- We present MultiLJP, the first real-world dataset for multi-defendant LJP, which facilitates future research in this area[1].
- We demonstrate the effectiveness of HRN on MultiLJP through empirical experiments.

---

[1] Our code and MultiLJP dataset are available at https://github.com/CURRENTF/HRN.

## 2 Related work

### 2.1 Legal judgment prediction

Legal judgment prediction has been studied in various jurisdictions (Zhong et al., 2020; Feng et al., 2022a; Katz et al., 2017; Chalkidis et al., 2019; Sulea et al., 2017a,b; Malik et al., 2021; Paul et al., 2020; Niklaus et al., 2021). Early studies on LJP focus on rule-based methods (Kort, 1957; Nagel, 1963; Segal, 1984) and machine learning algorithms (Aletras et al., 2016; Sulea et al., 2017a,b; Katz et al., 2017). Recent neural-based approaches jointly predict judgment results (law articles, charges and term of penalty) for single-defendant cases by modeling dependency between LJP subtasks (Zhong et al., 2018; Yang et al., 2019; Dong and Niu, 2021; Huang et al., 2021), leveraging legal knowledge (Hu et al., 2018; Gan et al., 2021; Yue et al., 2021; Ma et al., 2021; Lyu et al., 2022; Feng et al., 2022b), exploiting label information (Luo et al., 2017; Wang et al., 2019; Xu et al., 2020; Le et al., 2022; Liu et al., 2022; Zhang et al., 2023a), or employing pre-trained language models (Chalkidis et al., 2020, 2021; Xiao et al., 2021a). For multi-defendant cases, MAMD (Pan et al., 2019) utilizes multi-scale attention to distinguish confusing fact descriptions of different defendants for multi-defendant charge prediction.

However, existing single-defendant LJP methods neglect the complex interactions among multiple defendants. Moreover, compared with the multi-defendant LJP method MAMD (Pan et al., 2019), we follow the human judgment process to model hierarchical reasoning chains to distinguish different judgment results of various defendants and make accurate judgments for each defendant.

### 2.2 Multi-step reasoning with language models

Multi-step reasoning by training or fine-tuning language models to generate intermediate steps has been shown to improve performance (Zaidan et al., 2007; Talmor et al., 2020; Yao et al., 2021; Hase and Bansal, 2021; Zhang et al., 2023b; Gu et al., 2021). Ling et al. (2017) generate natural language intermediate steps to address math word problems. Camburu et al. (2018) extend the natural language inference dataset with human-annotated natural language explanations of the entailment relations. Rajani et al. (2019) generate rationales that explain model predictions for commonsense question-answering tasks (Talmor et al., 2019). Hendrycks

| MultiLJP | Number |
|---|---|
| # Training set cases | 18,968 |
| # Validation set cases | 2,379 |
| # Testing set cases | 2,370 |
| # Law articles | 22 |
| # Charges | 23 |
| # Terms of penalty | 11 |
| # Criminal relationships | 2 |
| # Sentencing circumstances | 8 |
| Total defendants | 80,477 |
| Average defendants | 3.39 |
| Law articles per defendant | 1.06 |
| Charges per defendant | 1.06 |
| Terms of penalty per defendant | 1 |
| Criminal relationships per defendant | 0.88 |
| Sentencing circumstances per defendant | 1.19 |
| Average length | 3,040.76 |

Table 1: Statistics of the MultiLJP.

et al. (2021) finetune pre-trained language models to solve competition mathematics problems by generating multi-step solutions. Nye et al. (2021) train language models to predict the final outputs of programs by predicting intermediate computational results. Recently, Wei et al. (2022) propose chain of thought prompting, which feed large language models with step-by-step reasoning examples without fine-tuning to improve model performance.

However, these methods are not designed for more realistic legal reasoning applications (Huang and Chang, 2022). In this paper, we aim to use generative language models to capture hierarchical reasoning chains for the multi-defendant LJP task.

## 3 Dataset

In this section, we describe the construction process and analyze various aspects of MultiLJP to provide a deeper understanding of the dataset.

### 3.1 Dataset construction

To the best of our knowledge, existing LJP datasets only focus on single-defendant cases or charge prediction for multi-defendant cases. Thus, we construct a Multi-defendant Legal Judgment Prediction (MultiLJP) dataset from the published legal documents in China Judgements Online[2]. Instead of extracting labels using regular expressions as in existing works (Xiao et al., 2018; Pan et al., 2019), we hire eight professional annotators to manually produce law articles, charges, terms of penalty, criminal relationships, and sentencing circumstances for each defendant in multi-defendant cases. The annotators are native Chinese speakers who have passed

---

[2] https://wenshu.court.gov.cn/

China's Unified Qualification Exam for Legal Professionals. All data is evaluated by two annotators repeatedly to eliminate bias. Since second-instance cases and retrial cases are too complicated, we only retain first-instance cases. Besides, we anonymize sensitive information (e.g., name, location, etc.) for multi-defendant cases to avoid potential risks of structured social biases (Pitoura et al., 2017) and protect personal privacy. After preprocessing and manual annotation, the MultiLJP consists of 23,717 multi-defendant cases. The statistics information of dataset MultiLJP can be found in Table 1.

## 3.2 Dataset analysis

**Analysis of the number of defendants**. MultiLJP only contains multi-defendant cases. The number of defendants per case is distributed as follows: 49.40 percent of cases have two defendants, 21.41 percent have three defendants, 11.22 percent have four defendants, and 17.97 percent have more than four defendants. The MultiLJP dataset has 80,477 defendants in total. On average, each multi-defendant case has 3.39 defendants.

**Analysis of multi-defendant judgment results**. In 89.58 percent of these cases, the defendants have different judgment results for at least one of the subtasks of the multi-defendant LJP task. Specifically, 18.91 percent of cases apply different law articles to different defendants, 26.80 percent of cases impose different charges on different defendants, and 88.54 percent of cases assign different terms of penalty to different defendants.

**Analysis of criminal relationships and sentencing circumstances**. Based on the gold labels of criminal relationships and sentencing circumstances, ideally, a judge can distinguish between 69.73 percent of defendants with different judgment results (law articles, charges, and terms of penalty). Specifically, based on the criminal relationship, a judge can distinguish 70.28 percent of defendants with different law articles and 72.50 percent of defendants with different charges; based on the sentencing circumstances, a judge can distinguish 96.28 percent of defendants with different terms of penalty.

## 4 Method

In this section, we describe the HRN method. First, we formulate our research problem. Then, we introduce the Sequence-to-Sequence (Seq2Seq) generation framework for hierarchical reasoning. Next, we introduce hierarchical reasoning chains of multi-defendant in detail. Finally, a training process with Fusion-in-Decoder for HRN is explained.

## 4.1 Problem formulation

We first formulate the multi-defendant LJP task. The fact description of a multi-defendant case can be seen as a word sequence $\mathbf{x} = \{w_1, w_2, ..., w_n\}$, where n represents the number of words. Each multi-defendant case has a set of defendant names $E = \{\mathbf{e}_1, \mathbf{e}_2, ..., \mathbf{e}_{|E|}\}$, where each name is a sequence of words $\mathbf{e} = \{w_1, w_2, .., w_{|\mathbf{e}|}\}$. Given the fact description $\mathbf{x}$ and the defendant name $\mathbf{e}$ of a multi-defendant case, the multi-defendant task aims to predict the judgment results of multiple applicable law articles, multiple charges, and a term of penalty. The law article prediction and the charge prediction subtasks are multi-label classification problems, and the term of penalty prediction subtask is a multi-class classification problem.

We introduce the criminal relationship and the sentencing circumstance as intermediate tasks to model the hierarchical reasoning chains for multi-defendant LJP and improve the prediction of the main judgment results. Criminal relationships refer to the relationships between defendants, specifically whether one defendant assisted other co-defendants during the commission of the crime. Sentencing circumstances refer to specific behaviors (such as confession and recidivist) or factors (like accessory and blind) that can influence the severity or leniency of a penalty[3]. These two tasks are also multi-label classification problems. We denote the labels of law articles, charges, terms of penalty, criminal relationships, and sentencing circumstances as word sequences $\mathbf{y}^l, \mathbf{y}^c, \mathbf{y}^t, \mathbf{y}^r$ and $\mathbf{y}^s$ respectively in this paper.

## 4.2 Sequence-to-sequence generation

From the perspective of Sequence-to-Sequence (Seq2Seq) generation, each task can be modeled as finding an optimal label sequence $\mathbf{y}$ that maximizes the conditional probability based on the fact description, a specific defendant name and a specific task description $p(\mathbf{y}|\mathbf{x}, \mathbf{e}, \mathbf{d})$, which is calculated

---

[3]Details of criminal relationships and sentencing circumstances definitions together with their explanations can be found in Appendix A and B.

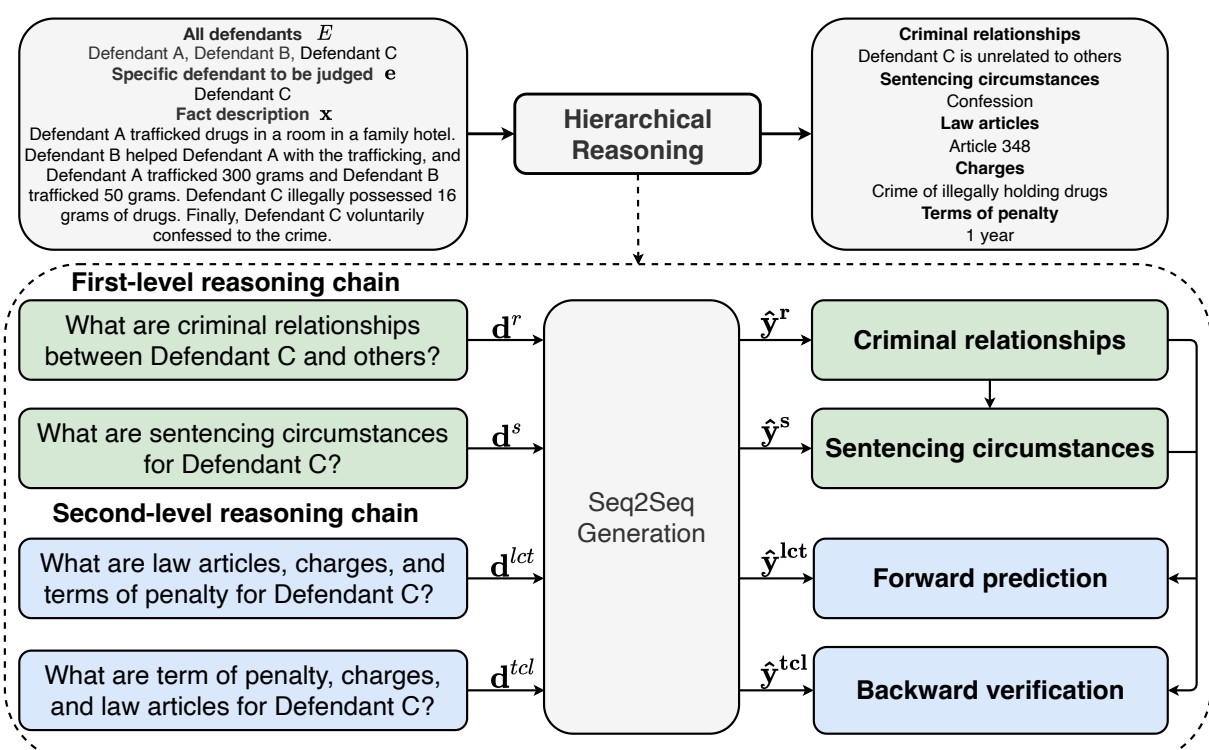

Figure 2: Overview of our proposed HRN. HRN leverages Sequence-to-Sequence (Seq2Seq) generation framework to follow hierarchical reasoning chains to generate prediction results.

as follows:

$$p(\mathbf{y}|\mathbf{x}, \mathbf{e}, \mathbf{d}) = \prod_{i=1}^{m} p(y_i|y_1, y_2, ..., y_{m-1}, \mathbf{x}, \mathbf{e}, \mathbf{d}),$$
(1)

where $m$ denotes the length of the label sequence, and the specific task description $\mathbf{d}$ is a semantic prompt that allows Seq2Seq generation models to execute the desired task. To accomplish the Seq2Seq generation tasks, we apply the Seq2Seq language model mT5 (Xue et al., 2021) to generate label sequences as follows:

$$\hat{\mathbf{y}} = DEC(ENC(\mathbf{x}, \mathbf{e}, \mathbf{d})),$$
(2)

where $ENC$ refers to the encoder of the language model, $DEC$ denotes the decoder of the language model, and $\hat{\mathbf{y}}$ is prediction results composed of words. We use special [SEP] tokens to separate the different information to form the input of the encoder.

### 4.3 Hierarchical reasoning chains

To distinguish different judgment results among various defendants, our method HRN follows hierarchical reasoning chains to determine each defendant's criminal relationships, sentencing circumstances, law articles, charges, and terms of penalty.

As shown in Figure 2, the hierarchical reasoning chains consist of two levels:

**The first-level reasoning is for intermediate tasks.** The first-level reasoning chain is to first identify relationships between defendants based on the fact description, the names of all defendants and the criminal relationship prediction task description $d^r$ as follows:

$$\hat{\mathbf{y}}^{\mathbf{r}} = DEC(ENC(\mathbf{x}, E, \mathbf{d}^r)).$$
(3)

Then, we determine sentencing circumstances for the defendant $\mathbf{e}$ based on the fact description, name of the defendant $\mathbf{e}$, prediction results of criminal relationships and the sentencing circumstance prediction task description $\mathbf{d}^s$, that is:

$$\hat{\mathbf{y}}^{\mathbf{s}} = DEC(ENC(\mathbf{x}, \mathbf{e}, \hat{\mathbf{y}}^r, \mathbf{d}^s)).$$
(4)

**The second-level reasoning is for judgment prediction tasks.** The second-level reasoning chain consists of a forward prediction process and a backward verification process. The forward prediction process is to predict law articles, charges, and terms of penalty (in that order) based on the fact description, the name of defendant $\mathbf{e}$, first-level reasoning results, and the forward prediction task description $\mathbf{d}^{lct}$ as follows:

$$\hat{\mathbf{y}}^{\mathbf{lct}} = DEC(ENC(\mathbf{x}, \mathbf{e}, \hat{\mathbf{y}}^r, \hat{\mathbf{y}}^s, \mathbf{d}^{lct})).$$
(5)

Then, the backward verification process is to verify these judgment results in reverse order based on the fact description, the name of defendant $\mathbf{e}$, first-level reasoning results and the backward verification task description $\mathbf{d}^{tcl}$, that is:

$$\hat{\mathbf{y}}^{\mathbf{tcl}} = DEC(ENC(\mathbf{x}, \mathbf{e}, \hat{\mathbf{y}}^r, \hat{\mathbf{y}}^s, \mathbf{d}^{tcl})). \quad (6)$$

### 4.4 Training with fusion-in-decoder

To handle multi-defendant fact descriptions whose average length exceeds the length limit of the encoder, we adopt Fusion-in-Decoder (FID) (Izacard and Grave, 2021) to encode multiple paragraphs split from a fact description. We first split the fact description $\mathbf{x}$ into $K$ paragraphs containing $M$ words. Then, we combine multiple paragraph representations from the encoder, the decoder generates prediction results by attending to multiple paragraph representations as follows:

$$\hat{\mathbf{y}} = DEC(\mathbf{h}_1, \mathbf{h}_2, ..., \mathbf{h}_K), \quad (7)$$

where $\mathbf{h}_i$ denotes the representation of the $i$-th paragraph of the fact description $\mathbf{x}$. Since all tasks are formulated as sequence-to-sequence generation tasks, we follow Raffel et al. (2020) to train the model by standard maximum likelihood and calculate the cross-entropy loss for each task. The overall loss function is formally computed as:

$$\mathcal{L} = \lambda_r \mathcal{L}_r + \lambda_s \mathcal{L}_s + \lambda_{lct} \mathcal{L}_{lct} + \lambda_{tcl} \mathcal{L}_{tcl}, \quad (8)$$

where hyperparameters $\lambda$ determine the trade-off between all subtask losses. $\mathcal{L}_r$, $\mathcal{L}_s$, $\mathcal{L}_{lct}$ and $\mathcal{L}_{tcl}$ denote the cross-entropy losses of the criminal relationship prediction task, the sentencing circumstance prediction task, the forward prediction process and the backward verification process, respectively. At test time, we apply greedy decoding to generate forward and backward prediction results. Finally, the chain with the highest confidence is chosen for the final prediction.

## 5 Experiments

### 5.1 Research questions

We aim to answer the following research questions with our experiments: (RQ1) How does our proposed method, HRN, perform on multi-defendant LJP cases? (RQ2) How do the different levels of reasoning chains affect the performances of HRN on multi-defendant LJP?

### 5.2 Baselines

To verify the effectiveness of our method HRN on multi-defendant LJP, we compare it with a variety of methods, which can be summarized in the following three groups:

- **Single-defendant LJP methods,** including **Top-judge** (Zhong et al., 2018), which is a topological dependency learning framework for single-defendant LJP and formalizes the explicit dependencies over subtasks as a directed acyclic graph; **MPBFN** (Yang et al., 2019), which is a single-defendant LJP method and utilizes forward and backward dependencies among multiple LJP subtasks; **LADAN** (Xu et al., 2020), which is a graph neural network based method that automatically captures subtle differences among confusing law articles; **NeurJudge** (Yue et al., 2021), which utilizes the results of intermediate subtasks to separate the fact statement into different circumstances and exploits them to make the predictions of other subtasks.

- **Pre-trained language models,** including **BERT** (Cui et al., 2021), which is a Transformer-based method that is pre-trained on Chinese Wikipedia documents; **mT5** (Xue et al., 2021), which is a multilingual model pre-trained by converting several language tasks into "text-to-text" tasks and pre-trained on Chinese datasets; **Lawformer** (Xiao et al., 2021b), which is a Transformer-based method that is pre-trained on large-scale Chinese legal long case documents.

- **Multi-defendant charge prediction method,** including **MAMD** (Pan et al., 2019), which is a multi-defendant charge prediction method that leverages multi-scale attention to recognize fact descriptions for different defendants.

We adapt single-defendant LJP methods to multi-defendant LJP by concatenating a defendant's name and a fact description as input and training models to predict judgment results. However, we exclude some state-of-the-art single-defendant approaches unsuitable for multi-defendant settings. Few-shot (Hu et al., 2018), EPM (Feng et al., 2022b), and CEEN (Lyu et al., 2022) annotate extra attributes for single-defendant datasets, not easily transferable to MultiLJP. Also, CTM (Liu et al., 2022) and CL4LJP (Zhang et al., 2023a) design specific sampling strategies for contrastive learning of single-defendant cases, hard to generalize to multi-defendant cases.

| Method | Law Articles | | | | Charges | | | | Term of Penalty | | | |
|---|---|---|---|---|---|---|---|---|---|---|---|---|
| | Acc. | MP | MR | F1 | Acc. | MP | MR | F1 | Acc. | MP | MR | F1 |
| TopJudge | 69.32 | 35.60 | 39.13 | 36.93 | 64.42 | 24.96 | 35.28 | 28.34 | 28.36 | 23.16 | 22.25 | 22.00 |
| MPBFN | 72.47 | 34.73 | 34.22 | 34.35 | 65.59 | 32.79 | 33.20 | 31.59 | 28.32 | 21.59 | 20.91 | 20.70 |
| LADAN | 54.57 | 38.09 | 22.40 | 26.64 | 46.62 | 20.68 | 32.42 | 24.74 | 27.05 | 24.05 | 23.43 | 23.16 |
| NeurJudge | 65.21 | 41.72 | 36.96 | 38.15 | 59.51 | 34.19 | 25.36 | 27.55 | 30.06 | 27.56 | 25.63 | 25.95 |
| BERT | 51.38 | 34.19 | 29.68 | 30.70 | 44.80 | 36.80 | 20.10 | 25.14 | 29.60 | 23.95 | 22.68 | 21.55 |
| mT5 | 87.49 | 74.28 | 53.65 | 58.84 | 81.52 | 63.33 | 49.94 | 52.86 | 33.66 | 39.13 | 24.23 | 23.04 |
| Lawformer | 75.50 | 36.18 | 35.33 | 34.00 | 65.94 | 38.97 | 29.12 | 32.76 | 32.37 | 22.66 | 20.68 | 18.30 |
| MAMD | - | - | - | - | 58.73 | 33.00 | 34.15 | 31.60 | - | - | - | - |
| **HRN** | **91.46**$^*$ | **69.87**$^*$ | **70.95**$^*$ | **69.20**$^*$ | **89.54**$^*$ | **71.80**$^*$ | **71.83**$^*$ | **70.70**$^*$ | **42.74**$^*$ | **41.33**$^*$ | **40.20**$^*$ | **40.62**$^*$ |
| w/ gold | 92.26$^*$ | 75.67$^*$ | 70.22$^*$ | 71.58$^*$ | 90.60$^*$ | 78.46$^*$ | 75.38$^*$ | 76.27$^*$ | 44.44$^*$ | 47.13$^*$ | 42.31$^*$ | 43.27$^*$ |

Table 2: Judgment prediction results on MultiLJP. Significant improvements over the best baseline are marked with $*$ (t-test, $p < 0.05$).

## 5.3 Implementation details

To accommodate the length of multi-defendant fact descriptions, we set the maximum fact length to 2304. Due to the constraints of the model input, BERT's input length is limited to 512. For training, we employed the AdamW (Loshchilov and Hutter, 2019) optimizer and used a linear learning rate schedule with warmup. The warmup ratio was set to 0.01, and the maximum learning rate was set to $1 \cdot 10^{-3}$. We set the batch size as 128 and adopt the gradient accumulation strategy. All models are trained for a maximum of 20 epochs. The model that performs best on the validation set is selected. For the hyperparameters, $\lambda$, in the loss function, the best setting is {0.6, 0.8, 1,4, 1.2} for {$\lambda_r$, $\lambda_s$, $\lambda_{lct}$, $\lambda_{tcl}$}. Additionally, we set the number of paragraphs $K$, the number of words per paragraph $M$, and the output length to 3, 768, and 64, respectively. For performance evaluation, we employ four metrics: accuracy (Acc.), Macro-Precision (MP), Macro-Recall (MR), and Macro-F1 (F1). All experiments are conducted on one RTX3090.

## 6 Experimental results and analysis

In this section, we first conduct multi-defendant legal judgment predictions and ablation studies to answer the research questions listed in the Section 5.1. In addition, we also conducted a case study to intuitively evaluate the importance of hierarchical reasoning.

### 6.1 Multi-defendant judgment results (RQ1)

Table 2 shows the evaluation results on the multi-defendant LJP subtasks. Generally, HRN achieves the best performance in terms of all metrics for all multi-defendant LJP subtasks. Based on the results, we have three main observations:

- Compared with state-of-art single-defendant LJP methods, e.g., Topjudge, MPBFN, LADAN, and NeurJudge, our method HRN consider hierarchical reasoning chains and thus achieve significant improvements. Since the single-defendant methods do not consider criminal relationships and sentencing circumstances, they can not distinguish different judgment results among various defendants well. It indicates the importance of following the hierarchical reasoning chains to predict criminal relationships and sentencing circumstances for multi-defendant LJP.

- As Table 2 shows, our method HRN achieves considerable performance improvements on all subtasks of multi-defendant LJP compared to pre-trained models BERT and Lawformer. This shows that modeling the hierarchical reasoning chains during the fine-tuning stage is crucial. Moreover, HRN, which combines mT5 and the hierarchical reasoning chains, significantly outperforms mT5, indicating that the language knowledge from pre-training and the information of the hierarchical reasoning chains are complementary to each other.

- Compared to MAMD designed for multi-defendant charge prediction, our method HRN outperforms MAMD on charge prediction task. This shows the effectiveness and robustness of modeling reasoning chains in real-world application scenarios.

- We employ gold annotations of criminal relationships and sentencing circumstances, rather than predicted ones, to enhance the performance of HRN. Our findings indicate a considerable improvement in results when relying on gold annotations. This observation underscores the potential for notable enhancement in multi-defendant

| Method | Law Articles | | | | Charges | | | | Term of Penalty | | | |
|--------|------|------|------|------|------|------|------|------|------|------|------|------|
| | Acc. | MP | MR | F1 | Acc. | MP | MR | F1 | Acc. | MP | MR | F1 |
| HRN | 91.46 | 69.87 | 70.95 | 69.20 | 89.54 | 71.80 | 71.83 | 70.70 | 42.74 | 41.33 | 40.20 | 40.62 |
| w/o CR | 89.14 | 73.71 | 67.85 | 68.25 | 86.84 | 72.81 | 68.61 | 68.95 | 37.83 | 38.20 | 30.30 | 30.69 |
| w/o SC | 88.58 | 73.64 | 61.51 | 65.51 | 85.98 | 77.03 | 64.82 | 68.98 | 32.66 | 37.37 | 25.43 | 24.56 |
| w/o FP | 89.05 | 73.61 | 65.61 | 68.17 | 85.97 | 73.24 | 66.09 | 68.29 | 35.83 | 34.35 | 30.47 | 30.00 |
| w/o BV | 90.91 | 75.60 | 67.58 | 68.50 | 83.45 | 69.34 | 60.86 | 63.73 | 37.41 | 36.90 | 28.42 | 28.88 |
| w/o all | 87.49 | 74.28 | 53.65 | 58.84 | 81.52 | 63.33 | 49.94 | 52.86 | 33.66 | 39.13 | 24.23 | 23.04 |

Table 3: Ablation studies on MultiLJP

**Multi-defendant fact description:**
Defendant A and defendant B were playing in a public place of entertainment when defendant A and the victim got into an argument and defendant B went up to see what was going on. Defendant A gathered defendant B and followed the victim to the entrance of the public place of entertainment, and defendant B came forward and beat the victim with fists and feet. The victim's injury constituted a second degree of minor injury.

**Gold labels for multiple defendants:**
**Defendant A:** law articles (article 293), charges (Crime of picking quarrels and provoking troubles), terms of penalty (15 months)
**Defendant B:** law articles (article 293), charges (Crime of picking quarrels and provoking troubles), terms of penalty (11 months)

**Prediction for multiple defendants (MAMD):**
**Defendant A:** law articles (article 264 ✗), charges (Theft ✗), terms of penalty (11 months ✗)
**Defendant B:** law articles (article 293 ✓), charges (Crime of picking quarrels and provoking troubles ✓), terms of penalty (11 months ✓)

**Prediction for multiple defendants (HRN):**
Criminal relationships: defendant B helped defendant A.
Sentencing circumstances of defendant A: none.
Sentencing circumstances of defendant B: accessory.
**Defendant A:** law articles (article 293 ✓), charges (Crime of picking quarrels and provoking troubles ✓), terms of penalty (15 months ✓)
**Defendant B:** law articles (article 293 ✓), charges (Crime of picking quarrels and provoking troubles ✓), terms of penalty (11 months ✓)

Figure 3: Case study for intuitive comparisons. Red and green represent incorrect and correct judgment results, respectively. Blue denotes descriptions of criminal relationships.

LJP by improving the first-level reasoning.

## 6.2 Ablation studies (RQ2)

To analyze the effect of the different levels of reasoning chains in HRN, we conduct an ablation study. Table 3 shows the results on MultiLJP with five settings: (i) w/o CR: HRN without predicting criminal relationships. (ii) w/o SC: HRN without predicting sentencing circumstances. (iii) w/o FP: HRN without the forward prediction process and predicts law articles, charges, and terms of penalty in reverse order. (iv) w/o BV: HRN without the backward verification process and predicts law articles, charges, and terms of penalty in order. (v) w/o all: HRN degrades to mT5 with multi-task settings by removing all reasoning chains.

Table 3 shows that all levels of reasoning chains help HRN as removing any of them decreases per-

formance:

- **Removing the first-level reasoning chain**. We observe that both criminal relationships and sentencing circumstances decrease the performance of HRN when we remove the first-level reasoning chains. Specifically, removing criminal relationships (CR) negatively impacts performance, especially on law articles and charges prediction, which means criminal relationships are helpful for distinguishing law articles and charges; removing sentencing circumstances (SC) negatively impacts performance, especially on terms of penalty, which means sentencing circumstances are helpful for distinguishing terms of penalty.

- **Removing the second-level reasoning chain**. We observe that the model without the second-level forward prediction process (FP) or the second-level backward verification process (BV) faces a huge performance degradation in multi-defendant LJP. As a result, although the model still performs the first-level reasoning, the absence of modeling forward or backward dependencies between LJP subtasks leads to poor LJP performances.

- **Removing all reasoning chains**. When removing all reasoning chains from HRN, there is a substantial drop in the performances of multi-defendant LJP. Experimental results prove that hierarchical reasoning chains can be critical for multi-defendant LJP.

## 6.3 Case study

We also conduct a case study to show how multi-defendant reasoning chains help the model distinguish judgment results of different defendants and make correct predictions. Figure 3 shows prediction results for two defendants, where red and green represent incorrect and correct predictions, respectively. Defendant B helped A beat the vic-

tim, but the fact description does not show a direct crime against the victim. Without determining criminal relationships and sentencing circumstances, MAMD focuses on the tailing behavior around A and misclassifies A's law articles, charges, and terms of penalty as article 264, theft, and 11 months. In contrast, by following reasoning chains to determine criminal relationships, sentencing circumstances, law articles, charges, and terms of penalty, HRN distinguishes different judgment results between defendants.

## 7    Conclusions

In this paper, we studied the legal judgment prediction problem for multi-defendant cases. We proposed the task of multi-defendant LJP to promote LJP systems from single-defendant to multi-defendant. To distinguish confusing judgment results of different defendants, we proposed a Hierarchical Reasoning Network (HRN) to determine criminal relationships, sentencing circumstances, law articles, charges and terms of penalty for each defendant. As there is no benchmark dataset for multi-defendant LJP, we have collected a real-world dataset MultiLJP. We conducted extensive experiments on the MultiLJP dataset. Experimental results have verified the effectiveness of our proposed method and HRN outperforms all baselines.

## Limitations

Although our work distinguishes the judgment results of multiple defendants by hierarchical reasoning, in real life, there exist many confusing charge pairs, such as (the crime of intentional injury, and the crime of intentional homicide). The fact descriptions of these confusing charge pairs are very similar, which makes it difficult for the multi-defendant LJP model to distinguish between confusing charge pairs. We leave this challenge for future work.

## Ethics Statement

Since multi-defendant legal judgment prediction is an emerging but sensitive technology, we would like to discuss ethical concerns of our work. Our proposed method HRN is a preliminary multi-defendant work and aims to assist legal professionals instead of replacing them. In addition, multi-defendant cases contain personal privacy information. To avoid potential risks of structured social biases (Pitoura et al., 2017; Lyu et al., 2023) and

protect personal privacy, we have anonymized sensitive information (e.g., name, location, etc.) for multi-defendant cases in MultiLJP dataset.

## Acknowledgments

This work was supported by the National Key R&D Program of China with grant No.2022YFC3303004, the Natural Science Foundation of China (62272274, 61902219, 61972234, 62102234, 62202271, 62072279, T2293773, 72371145), the Natural Science Foundation of Shandong Province (ZR2021QF129), the Key Scientific and Technological Innovation Program of Shandong Province (2019JZZY010129). All content represents the opinion of the authors, which is not necessarily shared or endorsed by their respective employers and/or sponsors.

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

## Appendix

## A  Criminal Relationship

Criminal relationships refer to the relationships between defendants, specifically whether one defendant assisted other co-defendants during the commission of the crime.

- **None**. One defendant did not help other defendants to conduct criminal behaviors.

- **Assistance**. One defendant helped other defendants to conduct criminal behaviors.

## B  Sentencing Circumstance

Sentencing circumstances refer to specific behaviors (such as confession and recidivist) or factors (like accessory and blind) that can influence the severity or leniency of a penalty.

- **Old People**. A person who has reached the age of 75 and conducts the crime is liable to a lighter punishment.

- **Deaf-mute or Blind**. A deaf-mute or blind person who conducts a crime is liable to a lighter punishment.

- **Accessory**. Accessory to the crime is liable to a lighter punishment.

- **Attempted Crime**. A person who has attempted to conduct a crime is liable to a lighter punishment.

- **Surrender**. A person who has surrendered to justice is liable to a lighter punishment.

- **Confession**. A person who has confessed to offence is liable to a lighter punishment.

- **Metitorious Serviceoffset**. A person who has metitorious serviceoffset is liable to a lighter punishment.

- **Recidivist**. Recidivist is liable to a heavier punishment.