# OpenReview forum: "Multi-Defendant Legal Judgment Prediction via Hierarchical Reasoning"
_EMNLP/2023/Conference — EMNLP 2023 Findings_

### Official Review · Reviewer_6vXn · 2023-07-23

**Soundness:** 4

**Excitement:**

4: Strong: This paper deepens the understanding of some phenomenon or lowers the barriers to an existing research direction.

**Paper Topic And Main Contributions:**

In this paper, the authors propose the task of multi-defendant LJP to promote LJP systems from single-defendant to multi-defendant. Specifically, they propose a Hierarchical Reasoning Network (HRN) to determine criminal relationships, sentencing circumstances, law articles, charges and terms of penalty for each defendant. Moreover, they provide the MultiLJP, the first real-world dataset for multi-defendant LJP tasks. Experimental results have verified the effectiveness of their proposed method.

**Reasons To Accept:**

1. The authors pay attention to the multi-defendant LJP tasks and provide the first real-world dataset for multi-defendant tasks.
2. The authors propose a Hierarchical Reasoning Network (HRN) to determine criminal relationships, sentencing circumstances, law articles, charges and terms of penalty for each defendant. Experimental results have verified the effectiveness of their proposed method.

**Reasons To Reject:**

1. The writing of this paper could be more clear. For example, the definition of criminal relationships and sentencing circumstances are not very clear (I can only guess the meaning from the example in Figure 1).
2. It seems the Forward prediction and Backward verification are very similar, the only difference is the order of those tasks. So this is more of a quadratic  Forward prediction process than a Backward verification.

**Reproducibility:**

4: Could mostly reproduce the results, but there may be some variation because of sample variance or minor variations in their interpretation of the protocol or method.

**Reviewer Confidence:**

5: Positive that my evaluation is correct. I read the paper very carefully and I am very familiar with related work.

---

> ### Author Rebuttal · Authors · 2023-08-28
>
> **Response to Reviewer 6vXn:**
>
> We sincerely thank you for your time and helpful feedback. We denote the Answer and Reason to reject as A and R respectively.
>
> **A1 to R.1:**
>
> Thanks for the advice. "Criminal relationships" refer to the relationships between defendants, specifically whether one defendant assisted other co-defendants during the commission of the crime. "Sentencing circumstances" refer to specific behaviors (such as confession and recidivism) or factors (like deafness and blindness) that can influence the severity or leniency of a penalty. We will provide clearer definitions of these terms in the paper.
>
> **A2 to R.2:**
>
> The motivation behind forward prediction and backward verification is rooted in the complex nature of legal reasoning, where evidence and conclusions can be interdependent. Unlike forward prediction, backward verification serves to examine the derived conclusions and backtrace the logic to ensure consistency. For example, the prediction of charges can confirm the rationality of the applicable laws. Our ablation experiments also demonstrate the effectiveness of both forward prediction and backward verification. We will further clarify the differences between forward prediction and backward verification in the paper.

---

### Official Review · Reviewer_udyE · 2023-07-29

**Paper Topic And Main Contributions:** 1. The authors propose a real-world m…
**Soundness:** 4

**Excitement:**

4: Strong: This paper deepens the understanding of some phenomenon or lowers the barriers to an existing research direction.

**Reasons To Accept:**

See details in Paper Topic And Main Contributions.

**Reasons To Reject:**

1. The authors propose a Hierarchical Reasoning process that uses multi-step reasoning to get the final judgment prediction. In order to implement the above method, I notice that the authors have labelled the data in detail, which is very commendable. In this regard, I have an idea that I would like to discuss with the authors. Specifically, since LLM has the CoT capability, I am curious if the CoT capability of LLM can be combined with the Hierarchical Reasoning approach in a judicial scenario. Or, whether LLM can be used as an annotator to annotate judicial data.

2. The datasets used by the authors are all from the China Judgements Online, but this is not consistent with the distribution of cases in real scenarios (since judgment documents are written by judges at the conclusion of a case). I am curious whether the author's method can work in daily counselling scenarios, maybe the author can show a few cases to solve my doubts.

3. If Criminal relationships and Sentencing circumstances are used as inputs (i.e., not predicted but inputted into the model along with the facts of the case), what would be the effect of the final judgement prediction?

4. Whether the Hierarchical Reasoning approach proposed by the authors can work on Single-Defendant Legal Judgment Prediction, i.e., removing the Criminal relationships prediction?

**Reproducibility:**

4: Could mostly reproduce the results, but there may be some variation because of sample variance or minor variations in their interpretation of the protocol or method.

**Reviewer Confidence:**

5: Positive that my evaluation is correct. I read the paper very carefully and I am very familiar with related work.

---

> ### Author Rebuttal · Authors · 2023-08-28
>
> **Response to Reviewer udyE:**
>
> We genuinely appreciate your valuable insights and the time you dedicated. We denote the Answer and Reason to reject as A and R respectively.
>
> **A1 to R.1:**
>
> The idea of integrating the CoT capability of LLMs with our hierarchical reasoning is intriguing. We anticipate that LLMs, enriched with legal knowledge and CoT capabilities, could be integrated seamlessly into our reasoning process. While LLMs could serve as potential annotators, there's a risk of mislabeling or introducing biases. Our MultiLJP dataset emphasizes accuracy and reduces biases via expert annotations and sensitive data removal. This dataset could guide the development of legal LLMs and help in the quality annotation of legal datasets using LLM's few-shot learning. We will explore the integration of the hierarchical reasoning process with legal LLMs in future work.
>
> **A2 to R.2:**
>
> Please note that the main purpose of our work is to assist judges in the task of multi-defendant legal judgment prediction and experimental results demonstrate the effectiveness of our method in this scenario. In daily counseling scenarios, users' descriptions are often brief and lack a complete description of the crime, making it difficult to accurately predict judgment results.  We will explore how to extend our method to daily counseling scenarios in future work.
>
> **A3 to R.3:**
>
> Thanks for the advice. We have conducted experiments using gold labels of criminal relationships and sentencing circumstances as input. As shown in the table below, considerably better results can be obtained when gold criminal relationships and sentencing circumstances labels are used. These results suggest that LJP results can be substantially improved by improving first-level reasoning (criminal relationships and sentencing circumstances). We will include these results in the paper.
>
> |  Method  | Law  Articles |       |       |       | Charges |       |       |       | Term  of  Penalty |       |       |       |
> |:--------:|:-------------:|:-----:|:-----:|:-----:|:-------:|:-----:|:-----:|:-----:|:-----------------:|:-----:|:-----:|:-----:|
> |          | Acc.          | MP    | MR    |   F1  |   Acc.  | MP    | MR    |   F1  |        Acc.       | MP    | MR    |   F1  |
> |    HRN   | 76.52         | 44.61 | 43.88 | 43.20 |  70.77  | 43.77 | 38.95 | 40.15 |       33.17       | 33.22 | 30.32 | 26.32 |
> | HRN+gold | 78.52         | 45.40 | 46.30 | 45.06 |  73.04  | 53.25 | 45.90 | 46.59 |       36.15       | 39.96 | 34.05 | 29.41 |
>
> **A4 to R.4:**
>
> Thanks for the suggestion. We have tested our HRN and baseline models on the single-defendant dataset CAIL-small. To ensure a fair comparison, all models are directly tested on the CAIL-small test set without fine-tuning on the CAIL-small training set. As shown in the table below, the experimental results demonstrate the effectiveness of HRN on single-defendant cases. We will include these results in the paper.
>
> |   Method  | Law  Articles |       |       |       | Charges |       |       |       | Term  of  Penalty |       |       |       |
> |:---------:|:-------------:|:-----:|:-----:|:-----:|:-------:|:-----:|:-----:|:-----:|:-----------------:|:-----:|:-----:|:-----:|
> |           |      Acc.     |   MP  |   MR  |   F1  |   Acc.  |   MP  |   MR  |   F1  |        Acc.       |   MP  |   MR  |   F1  |
> |  TopJudge |     40.71     | 21.18 | 24.09 | 21.14 |  40.31  | 26.60 | 25.02 | 21.98 |       18.61       | 11.37 | 12.40 | 11.02 |
> |   MPBFN   |     40.53     | 20.53 | 19.84 | 16.25 |  39.52  | 19.53 | 20.24 | 15.49 |       18.94       | 12.24 | 12.11 | 10.79 |
> |   LADAN   |     71.54     | 41.96 | 38.66 | 37.81 |  58.17  | 33.69 | 45.86 | 36.99 |       15.71       | 14.88 | 15.59 | 12.10 |
> | NeurJudge |     58.47     | 31.12 | 29.30 | 27.49 |  51.29  | 26.72 | 27.05 | 23.05 |       19.98       | 16.02 | 13.54 | 12.16 |
> |    BERT   |     63.10     | 25.20 | 21.36 | 21.75 |  60.66  | 24.45 | 18.95 | 20.05 |       16.02       | 12.04 | 14.49 |  6.92 |
> | Lawformer |     66.98     | 28.81 | 26.88 | 25.41 |  66.34  | 31.03 | 32.54 | 27.82 |       15.76       | 17.46 | 17.92 | 11.33 |
> |    MAMD   |       -       |   -   |   -   |   -   |  28.42  |  9.56 | 12.86 |  9.62 |         -         |   -   |   -   |   -   |
> |    HRN    |     72.04     | 54.68 | 52.02 | 50.11 |  70.51  | 54.41 | 52.38 | 50.38 |       21.93       | 18.04 | 12.11 | 10.37 |

---

### Official Review · Reviewer_TtcG · 2023-08-03

**Typos Grammar Style And Presentation Improvements:** 1.line 118, ‘mt5’ -> ‘mT5’.
2.line 36…
**Soundness:** 3

**Excitement:**

3: Ambivalent: It has merits (e.g., it reports state-of-the-art results, the idea is nice), but there are key weaknesses (e.g., it describes incremental work), and it can significantly benefit from another round of revision. However, I won't object to accepting it if my co-reviewers champion it.

**Paper Topic And Main Contributions:**

he paper extends LJP task from single-defendant to multi-defendant and proposes a multi-defendant LJP method called Hierarchical Reasoning Network (HRN). The proposed method employs Seq2Seq model and follows the hierarchical reasoning chains to generate prediction results. Moreover, this work releases a real multi-defendant LJP dataset. Results prove the effectiveness of HRN.

**Questions For The Authors:**

1.What is the motivation of forward and backward chains? Why use backward (tcl) in second-level reasoning chain? How about other combinations?

2.In the section 4.4, how to verify and get the final predictions in test set? What is the details after ‘greedy decoding’?

3.How to get the best setting of \lambda?

**Reasons To Accept:**

Strengths:
1.The trackled problem is attractive to the real work. The paper systematically analyzes the challenge of multi-defendant LJP task and further releases a dataset, which is beneficial to future work.

**Reasons To Reject:**

Weaknesses:
1.The writing could be improved further. The variable naming is confused, especially in Figure 2 without captions.

2.It is unclear that how to combine the inputs of each reasoning chain and how to utilize the outputs from prior reasoning chains. It is necessary to describe the detail task description in the section 4.3.

3.Multi-denfendant LJP task is extended from single-denfendant LJP task. So, single defendant can be regarded as a special situation of multiple defendant. The relationship of a single defendant could be “None” or ‘Single’. But in the experiments, this work only consider the multi-denfendant cases.



**Reproducibility:**

3: Could reproduce the results with some difficulty. The settings of parameters are underspecified or subjectively determined; the training/evaluation data are not widely available.

**Reviewer Confidence:**

4: Quite sure. I tried to check the important points carefully. It's unlikely, though conceivable, that I missed something that should affect my ratings.

---

> ### Author Rebuttal · Authors · 2023-08-29
>
> **Response to Reviewer TtcG:**
>
> We thank you for your efforts and valuable suggestions. We denote Answer, Reason to reject, Question and Typos as A, R, Q and T respectively.
>
> **A1 to R.1:**
>
> We will clarify the variable names in the caption of Figure 2.
>
> **A2 to R.2:**
>
> To clarify, we employ special [SEP] tokens to differentiate various text segments and concatenate them for encoder input. Outputs from the first-level reasoning chain are fed into the second-level reasoning chain. We'll provide an expanded explanation in section 4.3.
>
> **A3 to R.3:**
>
> Thanks for the suggestion, we have evaluated our HRN and baselines on the single-defendant dataset CAIL-small. To ensure a fair comparison, all models are directly tested on the CAIL-small test set without fine-tuning on the CAIL-small training set. As shown in the table below, the experimental results demonstrate the effectiveness of HRN on single-defendant cases. We will include these results in the paper.
>
> |   Method  | Law  Articles |       |       |       | Charges |       |       |       | Term  of  Penalty |       |       |       |
> |:---------:|:-------------:|:-----:|:-----:|:-----:|:-------:|:-----:|:-----:|:-----:|:-----------------:|:-----:|:-----:|:-----:|
> |           |      Acc.     |   MP  |   MR  |   F1  |   Acc.  |   MP  |   MR  |   F1  |        Acc.       |   MP  |   MR  |   F1  |
> |  TopJudge |     40.71     | 21.18 | 24.09 | 21.14 |  40.31  | 26.60 | 25.02 | 21.98 |       18.61       | 11.37 | 12.40 | 11.02 |
> |   MPBFN   |     40.53     | 20.53 | 19.84 | 16.25 |  39.52  | 19.53 | 20.24 | 15.49 |       18.94       | 12.24 | 12.11 | 10.79 |
> |   LADAN   |     71.54     | 41.96 | 38.66 | 37.81 |  58.17  | 33.69 | 45.86 | 36.99 |       15.71       | 14.88 | 15.59 | 12.10 |
> | NeurJudge |     58.47     | 31.12 | 29.30 | 27.49 |  51.29  | 26.72 | 27.05 | 23.05 |       19.98       | 16.02 | 13.54 | 12.16 |
> |    BERT   |     63.10     | 25.20 | 21.36 | 21.75 |  60.66  | 24.45 | 18.95 | 20.05 |       16.02       | 12.04 | 14.49 |  6.92 |
> | Lawformer |     66.98     | 28.81 | 26.88 | 25.41 |  66.34  | 31.03 | 32.54 | 27.82 |       15.76       | 17.46 | 17.92 | 11.33 |
> |    MAMD   |       -       |   -   |   -   |   -   |  28.42  |  9.56 | 12.86 |  9.62 |         -         |   -   |   -   |   -   |
> |    HRN    |     72.04     | 54.68 | 52.02 | 50.11 |  70.51  | 54.41 | 52.38 | 50.38 |       21.93       | 18.04 | 12.11 | 10.37 |
>
> **A4 to Q.1:**
>
> The motivation behind employing both forward and backward chains is rooted in the complex nature of legal reasoning, where evidence and conclusions can be interdependent. The backward chain (tcl) in the second-level reasoning chain serves to examine the derived conclusions and backtrace the logic to ensure consistency.  For example, the prediction of the charges can verify the rationality of predicted applicable laws. Our ablation experiments have demonstrated the effectiveness of the backward chain, and we would like to explore other combinations in future work.
>
> **A5 to Q.2:**
>
> After implementing 'greedy decoding', we aggregate the predictions from various reasoning chains. Specifically, the chain with the highest confidence is chosen for the final prediction. We will add these details in section 4.4.
>
> **A6 to Q.3:**
>
> We conducted a grid search on the validation set to determine the best setting for \lambda. We will include this in the paper.
>
> **A7 to T.1&T.2&T.3:**
>
> We will fix these typos.

---

### Meta-Review · Area_Chair_EFzA · 2023-09-17

**Recommendation:** 3

**Metareview:**

The paper considers the task of of multi-defendant legal judgement prediction (LJP) task. This is an under-explored area in LJP literature. The authors introduce a new dataset resource and modelling approach. The dataset introduced in this paper contains a substantial number of instances based on real-world cases in China.  The modelling approach and baseline applies fusion in decoder architecture seq2seq model for an interesting hierarchical approach for extracting facts and relations from the case data.

There is reasonable agreement between the reviewers about the strength and excitement of this work and most questions seem to be well addressed by the authors. However, the reasons to accept listed by the reviewers seem to be fairly surface level. I think there are merits to the work, but the contributions are mostly collecting open data and applying a model architecture in an interesting way. I think the work has the potential to be used by other researchers studying LJP.

---

### Decision · Program_Chairs · 2023-10-07

**Decision:**

Accept-Findings

**Comment:**

The paper considers the task of of multi-defendant legal judgement prediction (LJP) task. This is an under-explored area in LJP literature. The authors introduce a new dataset resource and modelling approach. The dataset introduced in this paper contains a substantial number of instances based on real-world cases in China.  The modelling approach and baseline applies fusion in decoder architecture seq2seq model for an interesting hierarchical approach for extracting facts and relations from the case data.

There is reasonable agreement between the reviewers about the strength and excitement of this work and most questions seem to be well addressed by the authors. However, the reasons to accept listed by the reviewers seem to be fairly surface level. I think there are merits to the work, but the contributions are mostly collecting open data and applying a model architecture in an interesting way. I think the work has the potential to be used by other researchers studying LJP.